# Reliability of Isometric Muscle Strength Measurement and Its Accuracy Prediction of Maximal Dynamic Force in People with Multiple Sclerosis

**DOI:** 10.3390/medicina58070948

**Published:** 2022-07-18

**Authors:** Kora Portilla-Cueto, Carlos Medina-Pérez, Ena Monserrat Romero-Pérez, José Aldo Hernández-Murúa, Carolina Vila-Chã, José Antonio de Paz

**Affiliations:** 1Institute of Biomedicine (IBIOMED), University of León, 24071 León, Spain; cora1995port@gmail.com (K.P.-C.); carlosmedinaper85@gmail.com (C.M.-P.); 2Division of Biological Sciences and Health, University of Sonora, Hermosillo 83000, Mexico; 3Escuela Superior de Educación Física, Universidad Autónoma de Sinaloa, Culiacán 80040, Mexico; aldohdez80@hotmail.com; 4Research Center in Sports Sciences, Health and Human Development (CIDESD), 5001-801 Vila Real, Portugal; cvilacha@ipg.pt

**Keywords:** multiple sclerosis, reliability, 1RM, maximum voluntary isometric contraction, minimum detectable change

## Abstract

*Background and Objectives*: Multiple sclerosis (MS) is a disease that manifests with varied neurological symptoms, including muscle weakness, especially in the lower extremities. Strength exercises play an important role in the rehabilitation and functional maintenance of these patients. The individualized prescription of strength exercises is recommended to be based on the maximum force determined by the one-repetition maximum (1RM), although to save time and because it requires less equipment, it is often determined by the maximum voluntary isometric contraction (MVIC). The purpose of this work was to study, in patients with MS (pwMS), the reliability of MVIC and the correlation between the MVIC and 1RM of the knee extensors and to predict the MVIC-based 1RM. *Materials and Methods**:* A total of 328 pwMS participated. The study of the reliability of MVIC included all pwMS, for which MVIC was determined twice in one session. Their 1RM was also evaluated. The sample was randomized by MS type, sex, and neurological disability score into a training group and a testing group for the analysis of the correlation and prediction of MVIC-based 1RM. *Results*: MVIC repeatability (ICC, 2.1 = 0.973) was determined, along with a minimum detectable change of 13.2 kg. The correlation between MVIC and 1RM was R^2^ = 0.804, with a standard error estimate of 12.2 kg. The absolute percentage error of 1RM prediction based on MVIC in the test group was 12.7%, independent of MS type and with no correlation with neurological disability score. *Conclusions:* In patients with MS, MVIC presents very good intrasubject repeatability, and the difference between two measurements of the same subject must differ by 17% to be considered a true change in MVIC. There is a high correlation between MVIC and 1RM, which allows estimation of 1RM once MVIC is known, with an estimation error of about 12%, regardless of sex or type of MS, and regardless of the degree of neurological disability.

## 1. Introduction

Multiple sclerosis (MS) is a chronic inflammatory disease of the central nervous system characterized by axonal demyelination [1] and formation of gliotic scars in inactive lesions [2]. It is considered the most frequent non-traumatic disabling disease in young adults [3], with a prevalence that is three times higher in women than in men [4]. Throughout its chronic course, it can manifest with both physical and mental symptoms and with irreversible neurological deficits such as muscle weakness, spasticity, or fatigue, among others [5].

Classically, three evolutionary forms are recognized: relapsing-remitting MS (RRMS), where exacerbations occur over time and are followed by total or partial functional recovery; secondary progressive MS (SPMS), characterized by evolving without exacerbations but gradually accumulating functional deficits; and primary progressive MS (PPMS), characterized by progressive functional deterioration from the onset of the disease, without ever presenting phases of exacerbation and remission [6].

The functional symptoms that patients with multiple sclerosis (pwMS) usually present with, along with a usual lower level of physical activity compared to healthy controls [7], contribute to increased disability and poorer quality of life [8,9].

Until recently, it was believed that physical exercise could worsen or accelerate the symptoms of the disease, which is why its prescription was avoided. However, physical exercise is currently considered safe for pwMS [9] and is a fundamental part of its non-pharmacological treatment. Although exercise does not modify the clinical course of the disease, it can help mitigate some of its symptoms [10,11].

Muscle strength is an important health-related factor, both for facilitating functional independence [12] and because it is associated with a lower risk of premature all-cause mortality [13].

PwMS tend to have lower muscle performance than healthy subjects of the same age and sex [7,14], mainly in the lower extremities [7]. This negatively affects the performance of everyday tasks, such as climbing stairs, sitting down, and getting up [7]. Currently, pwMS are advised to carry out the exercise, especially strength training, due to the specific benefits it has in terms of functional capacity, decreased fatigue, and increased walking ability [15,16].

Most of the published consensuses in relation to the prescription of strength exercises for health advise individualizing the workload based on the patient’s maximum strength, understood as the maximum load that they are capable of mobilizing once correctly (the one-repetition maximum, or 1RM) [17,18]. However, until relatively recently, the fear of performing a 1RM assessment due to the preconception that injuries could be caused by an inadequate execution technique [19], alongside doubt about the viability of this evaluation in pwMS due to the habitual fatigue that is often present, meant that the maximum isometric force (maximum voluntary isometric contraction, or MVIC) was used as a reference for programming the load in pwMS, instead of the 1RM [20,21,22,23], and even as outcome variable for the study of strength training in these patients. When a result variable is used (for example, the MVIC) to assess the effect of an intervention (for example, exercise), it is important to determine the degree of agreement between the values obtained in a repeated evaluation in the same subject. This allows us to obtain the SEM and MDC of the method so that we can make a more accurate interpretation of our results because if the magnitude of the change experienced as a consequence of the intervention is greater than the MDC, we will have greater certainty that the change is not due to a lack of precision in the method used.

Isometric maximal strength has high test–retest reliability in different healthy populations of varied physical conditions [24,25,26,27], but in pwMS, this relationship has not yet been published in widely distributed scientific journals.

MVIC is used with some frequency as an exercise model in fatigue studies or in electrophysiological studies [28,29,30,31,32,33], while 1RM is currently mostly used in studies in which resistance exercises are performed as a component of rehabilitation treatment [15,34,35,36]. 

In order to make comparative approximations between the pwMS participating in studies that measure these two manifestations of force in a dichotomous way, it is pertinent to study the correlation between them.

The purpose of this study was, in patients with MS, to analyze the absolute reliability of the assessment of the maximum isometric force of knee extension and to analyze the correlation and the prognostic value that the MVIC has in the prediction of dynamic maximal strength (assessed with 1RM).

## 2. Materials and Methods

### 2.1. Study Design

To study the reliability of determining the isometric strength, MVIC was evaluated for 328 pwMS, in the same session, on two occasions separated by five minutes.

For the analysis of the correlation between the MVIC and the 1RM and the study of the predictive value of the first over the second, cross-validation was carried out in which 328 pwMS were stratified randomly by sex, type of MS, and degree of disability into the study group (training group) and the verification group (testing group). This design is shown in Figure 1:

### 2.2. Population and Sample

This observational cross-sectional study was part of a strength training service for PwMS, which was performed by the University of León with the support of the Regional Ministry of Health of the Government of Castilla y León (Spain). All 328 participants (123 men and 205 women) were people affected by multiple sclerosis belonging to one of the nine MS patient associations of Castilla y León and diagnosed by the neurology medical services of the Regional Health System of each of the cities where the associations are located. All evaluations were conducted by the same researchers from the Exercise Physiology Laboratory at the University of Leon between March 2019 and March 2022. A doctor collected the medical history of each patient and also evaluated the degree of neurological disability based on the Expanded Disability Status Scale (EDSS) [37]. The different levels of disability were taken from previous similar works [38,39], considering mild = EDSS ≤ 2.5; moderate = EDSS ≤ 5; severe = EDSS ≤ 7.5; or very severe = EDSS > 7.5.

The inclusion criteria were as follows: confirmed diagnosis of MS based on the criteria established by McDonald (2001) [40], the ability to walk with or without a cane, relapse-free and without evident functional worsening in the previous 60 days, and with the ability to perform strength tests. Those cases in which this type of effort was contraindicated by the responsible physician due to acute health problems or other clinically uncontrolled comorbidities were excluded.

Participation was voluntary and consented to in writing. The research protocol was approved by the Research Ethics Committee of the University of León (30 January 2017, study number 1835). The methods, procedures and data processing were carried out in accordance with the relevant ethical standards and the Declaration of Helsinki (revised in October 2013).

For the cross-validation study, 328 patients were ordered in a spreadsheet based on three criteria: first criterion: type of sclerosis (three types), second criterion: sex (two types), and third criterion: degree of disability (four degrees). Once all the participants were ordered, a consecutive sequence of numbers was generated. All those with odd numbers formed the study group, and all those with even numbers the verification group.

### 2.3. Measurements

MVIC and 1RM were measured in the knee extensors using a multi-station machine (BH^®^ fitness Nevada Pro-T, Madrid, Spain).

The length of the lever arm was adjusted according to the leg length of each patient so that the area of force application was slightly above the tibia–fibula malleolar axis.

#### 2.3.1. Maximum Voluntary Isometric Contraction (MVIC)

The MVIC of the knee extensors was measured with a load cell (Globus, Codogné, Italy; with a sampling frequency of 1000 Hz) and its associated software (Globus Ergo Tester v1.5, Codogné, Italy), similarly to methods described in previous works [20,21,41], sitting with a hip flexion of 110° and knee flexion between 90 and 95°, and measured with a TEC goniometer (Sport-Tec Physio & Fitness, Pirmasens, Germany). Patients were instructed to push as hard as possible from the start of the test and to maintain that tension for five seconds. For the correlation study with the 1RM, the highest maximum value of two isometric attempts was used. All participants received verbal reinforcement during the effort.

#### 2.3.2. One-Repetition Maximum (1RM)

One-repetition maximum assessment of the knee extensors was performed ten minutes after the MVIC test, following previously published protocols [39,41]. First, four repetitions were performed at 50% MVIC. After each set, the patient reported subjective perceived exertion (RPE) via the OMNI-RES Endurance Exercise Scale [42]. After this, a series of two repetitions were performed, with a two-minute rest between each series until reaching the 1RM, which was determined in no more than six series. If it was not possible to determine 1RM with a maximum of six series, the determination was repeated again 48 h later. The load was increased by between 5 and 14 kg depending on the RPE and the quality of the execution technique, and when the patient could not lift the load even once, a lower weight was put on, in approximations of 2.5 kg.

### 2.4. Data Analysis

Descriptive analysis of the quantitative variables is presented as the mean and standard deviation (SD), and qualitative variables are shown as counts and percentages. The normality of the distribution of the variables was determined using the Kolmogorov–Smirnov test. For the analysis of the reliability of the determination of the MVIC, we considered the coefficient of variation, the intraclass correlation quotient (ICC) [1,2] for absolute agreement of single measurements, the standard error of the measurement (SEM) (SEM = SD × √ (1−ICC)), and the minimum detectable change (MDC) (MDC95 = SEM × 1.96 × √2). The correlation analysis between MVIC and 1RM and the function that describes their relationship was carried out with a simple linear regression analysis. A comparison of quantitative variables between patients with different types of MS or with different degrees of EDSS was performed with a one-way ANOVA test with post hoc Bonferroni tests, when relevant; comparison between sexes was through Student’s *t*-test of independent samples, and the effect size was calculated with Cohen’s d (d). The study of the association between qualitative variables was carried out using the chi-squared test. A comparison between the measured 1RM and the estimated 1RM with two different equations was performed with an ANOVA of repeated measures, and the effect size was calculated with the partial eta-squared (η^2^). G*Power software version 3.1.9.7, (Heinrich-Heine-University, Düsseldorf, Germany), was used to calculate the minimum required sample size and the calculated power. To detect a minimum effect size of 0.2, with an alpha error of 0.05 and a power of 0.95 with two tails using a single predictor, a minimum sample size of 67 was calculated. After finding the correlation between the predictor variable (isometric strength) and the outcome variable (1RM) obtained in half of our sample, the post hoc power calculated with the G*Power software (version 3.1.9.7) was 1.0 (100%). A minimum significance level of *p* < 0.05 was established. All the tests were carried out with the statistical program SPSS 26.0 for Mac (IBM Inc., Chicago, IL, USA).

## 3. Results

Table 1 shows the degree of neurological disability (EDSS) and the types of MS presented by the study participants.

There is an association between the type of MS and the degree of EDSS (χ2 (8, N = 328) = 57.14, *p* < 0.001), with a lower proportion of mild EDSS in RRMS and a higher proportion of severe EDSS in SPMS.

The descriptive values of age, BMI, years of evolution, EDSS, MVIC, and 1RM of the sample, separated by sex, are shown in Table 2.

Between the men and women of the sample, there were only significant differences (*p* < 0.001) in the strength values (MVIC and 1RM), with a large effect size (1.301 and 1.147, respectively).

Table 3 shows the values relative to the absolute reliability (ICC, SEM, CV, and MDC) of the determination of the maximum isometric force (MVIC) of the knee extensor muscles.

High reliability was found for an ICC measure of absolute agreement (0.973), with a CV of 4.5%, an SEM of 4.7 kg (6.3%), and an MDC of 13.2 kg.

The values of age, BMI, years of evolution, EDSS, MVIC, and 1RM of the sample, separated by training group and testing group, used for cross-validation of the correlation between MVIC and 1RM, are shown in Table 4.

We did not find significant differences in the variables shown between the two groups of the cross-validation study.

The parameters that define the linear regression function between MVIC (independent variable) and 1RM (dependent variable) obtained in the entire training group, and also those obtained separately in the men and women of this group, are presented in Table 5, considering that the equation of the linear regression is defined as y = ax + b, with “y” being the dependent variable, “x” the independent variable, “a” the slope of the line, and “b” the intercept (value of “y” when “x” = 0).

The correlation obtained between the two expressions of strength measured, both in the complete subsample of the training group and separately in that of men or women of this group, was high (R = 0.897, R = 0.854, and R = 0.895, respectively), and each increase of 1 kg in the isometric force was associated with an increase of 0.833 kg in the 1RM in the entire sample. The values were 0.718 kg in men and 1.024 kg in women, with a high value of the coefficient of determination at R^2^ = 0.804, R^2^ = 0.729, and R^2^ = 0.801, respectively. The standard error of the estimate (SEE) shows that the average difference between the measured value and the one estimated with the general equation is 12.2 kg, while the estimate with the regression by sex is 14.8 kg in men and 9.4 kg in women.

Table 6 shows the values of MVIC and 1RM for the testing group, as well as the predictions of 1RM from the MVIC when applying the regression functions between these variables obtained in all training groups and those obtained for each of the sexes and the percentage differences between the measured and estimated 1RM.

When applying the correlation equation between 1RM and MVIC found in the training group to the subjects of the testing group, it was observed that there were no significant differences between the estimated (1RM_all) and the measured (1RM) values, despite the fact that there was an error absolute average in the estimation of 12.7%.

When comparing the 1RM with the 1RM estimated from the regression equations obtained in the entire training group (1RM_all) and the values obtained separately for each of the sexes (1RM_sex), no differences were observed between the three values in men (*p* = 0.271) or in women (*p* = 0.731). We also found no differences between the absolute percentage errors of the estimation of the 1RM_all and 1RM_sex with respect to the 1RM: *p* = 0.963 in men and *p* = 0.589 in women.

Figure 2 shows the correlation between the degree of neurological disability (EDSS) and the absolute percentage difference in the estimation of the 1RM_all with respect to the 1RM, a correlation that was not significant.

Table 7 shows the absolute percentage differences in the estimation between the 1RM and 1RM_all in the patients of different evolutionary groups of the disease, for which it was shown that there were no differences in the percentage estimation errors of the 1RM_all (F (2.163) = 0.791, *p* = 0.456).

## 4. Discussion

In the present study, we found that the assessment of the MVIC of the extensors of the knee in pwMS shows excellent repeatability. In addition, we confirmed a high correlation of MVIC with 1RM, and in the cross-validation study, we also found that MVIC has a high predictive value for 1RM.

Currently, the importance of maintaining a good level of muscle strength is recognized in order to maintain functional independence in people whose health may be threatened (for example, in the elderly [43,44,45]). Its role in maintaining functional capacity has also been recognized; for example, in patients with COPD [46,47], heart patients [48], and diabetics [49,50]. The WHO’s recommendations for physical activity for the general population include physical exercise for muscle strengthening at least twice a week [51].

In general, rehabilitation programs for patients with any disease include a strength training component, and among the objectives pursued in general is achieving a gain in maximum dynamic strength [52,53,54].

Taking into account the relatively low prevalence of this disease in countries such as Spain, with an approximate ratio of 100 cases per 100,000 inhabitants [55], the number of pwMS that participated in this study represents a large sample compared to the sample size of published muscle training studies conducted with pwMS [56]. The sample consisted of a greater proportion of women (62.5%), as expected, since the prevalence in women is higher [57]. The mean age of our sample was 46.5 (±11.3) years, which is normal considering that the average evolution time is 10 years and that the disease typically begins in young people [3]. The most frequent evolutionary form of MS in the sample subjects was RRMS (65.8%), which is the most common type of MS globally [58].

An important clinical parameter used to assess the degree of neurological disability caused by the disease is the use of EDSS, and 48% of the sample had a moderate degree of neurological disability. We found a significant association between the type of MS and the degree of EDSS (χ^2^ (8, N = 328) = 57.14, *p* < 0.001), with the patients with RRMS presenting a lower degree of neurological involvement (84%, moderate or mild), and patients with SPMS presenting a higher proportion of neurological involvement (58% severe or very severe neurological disability). These differences can be explained because SPMS is a form of disease progression that is preceded by the RRMS form [59].

There was no difference between male and female patients in terms of age, BMI, years of disease progression, or degree of disability, although female patients had a lower degree of isometric strength and 1RM, as expected [39,60].

It is very important to know the SEM and MDC of the methods used in the quantification of a variable result; if the observed change is not greater than the MDC, regardless of the statistical significance found, we will have a reasonable doubt as to whether this change is real or whether it may be due to the random error of the measurement method [61,62]. 

The repeatability of the MVIC evaluation was very high (ICC (2,1) 0.973), with an SEM of 4.7 kg, expressed as a percentage of 6.3% of the measurement. Likewise, the MDC for this method was 13.2 kg (17.4%). This is an important piece of information that too often is not considered before drawing conclusions about the effectiveness of a training program [63,64].

Due to fear, lack of time, or insufficient materials for the measurement of 1RM, with some frequency, the maximum force is estimated from the MVIC since it is known that there is a good correlation in general between the two types of force. However, the fact that the correlation is assumed to be good does not allow us to be confident of the error in the estimation unless this error has been previously studied. 

The correlation was very good between both forces (R = 0.897) in this study, and the degree of correlation did not improve when analyzed separately in men and women, which allowed us to use a single correlation equation regardless of the patient’s sex. We did need to take into account that the specific equation for men entails a greater error in the estimation of the 1RM (around 2.6 kg more error) than the generic equation, while the specific equation for women produces a smaller error in the estimate than if the generic equation is used—about 5.4 kg less.

To study the performance of the generalization of the correlation that we found in pwMS between the MVIC and 1RM, we carried out a cross-validation study [65] with a randomized sample so that the number of patients, the type of MS, and the value of EDSS were similar in the training group and the testing group. 

We applied the correlation found in the training group to the testing group. We did not find significant differences between the measured value of 1RM and the estimated value when applying the regression equation (*p* = 0.463), although, on average, the error in the estimate was around 12.7%. Part of this difference is attributable to the fact that the approximation of 1RM that we made was 2.5 kg (minimum interval between the next two weights), and since the number of series for its determination is recommended to be a maximum of 6 [48], and considering that the average 1RM of the sample was 72.6 kg, at least 3.4% error is attributable to the degree of discrimination of the method for close loads.

Another noteworthy aspect of our results is that, despite the fact that the error in the estimation of 1RM in women tends to be lower when a specific correlation equation is used, the use of a generalized equation (regardless of sex) does not produce significant differences.

However, there are several evolutionary forms of the disease, so we compared the errors obtained in the estimation of 1RM between patients with different types of MS. It is noteworthy that the error in the estimation was similar between patients with RRMS, SPMS, and PPMS, so the same equation is usable regardless of the type of MS.

The difference between patients with MS does not lie only in the type of MS they present but also in the degree of impairment they suffer in their nervous system. The most common way to assess the degree of disability is the use of the EDSS [66,67], and the degree of EDSS has been reported to negatively correlate with a functional capacity [68]. For this reason, we also analyzed whether the error in the estimation of the 1RM was correlated with the EDSS, and we found that there was no significant correlation between the error and the 1RM (Figure 2).

We believe that the present study is of practical interest to professionals who are dedicated to the rehabilitation or physical reconditioning of patients with MS since it provides information on the inherent error and the minimum detectable change in the evaluation of the isometric strength of the knee extensors in these patients.

In addition, knowledge of the existing correlation between the MVIC and 1RM allows these values to be estimated in studies in which only one of the two muscle forces was analyzed. 

However, there are some limitations to this study. The first is that the isometric force produced is related to the joint angle of the exercise [69,70]. Therefore, this extrapolation is valid only when the knee is flexed at around 95° and the hip at 110°. The results for weight machines with different angles between their levers can produce greater errors when applying the regression equation of this study.

Moreover, cross-sectional observational studies, such as the present study, do not report causal relationships or the effects of interventions. Therefore, we do not know whether the correlation found between MVIC and 1RM would be maintained or not after a period of strength training in these patients. Future studies will be needed to analyze the evolution of this correlation over a period of training. It will also be interesting to analyze and compare the strength values and correlations between these manifestations of muscle force in patients with similar neurological damage determined by magnetic resonance imaging and functional magnetic resonance imaging.

We should emphasize that one of the strengths of the present study is the size of the sample, as well as the cross-validation by dividing with the division of the sample into a training group and a verification group so that the predictive value of the MVIC on the 1RM could be extrapolated samples with similar characteristics. In addition, this study provides initial information on the repeatability of the MVIC and the predictive value of the 1RM in patients with multiple sclerosis, aspects not published in scientific journals to date.

## 5. Conclusions

In patients with MS, the MVIC has a very good intra-subject repeatability, and the difference between two measurements of the same subject must differ by 17% to be considered a true change MVIC. There is a high correlation between MVIC and 1RM, which allows us to estimate the 1RM once the MVIC is known, with an estimation error of around 12% regardless of sex or type of MS and regardless of the degree of EDSS.

## 6. Ethical Aspects 

The research protocol was approved by the Research Ethics Committee of the University of León (30 January 2017, study number 1835). The methods, procedures, and data processing were carried out in accordance with the relevant ethical standards and the Declaration of Helsinki (revised in October 2013). In addition, the Spanish legislation in force in the field of biomedical research (Law 14/2007, of 3 July, on Biomedical Research and Law 41/2002, of 14 November, on patient autonomy and rights and obligations regarding clinical information and documentation) and on personal data protection were followed.

## Figures and Tables

**Figure 1 medicina-58-00948-f001:**
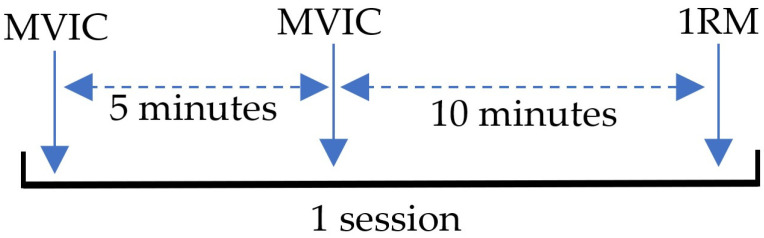
Evaluation design. 1RM—one-repetition maximum; MVIC—maximum voluntary isometric contraction.

**Figure 2 medicina-58-00948-f002:**
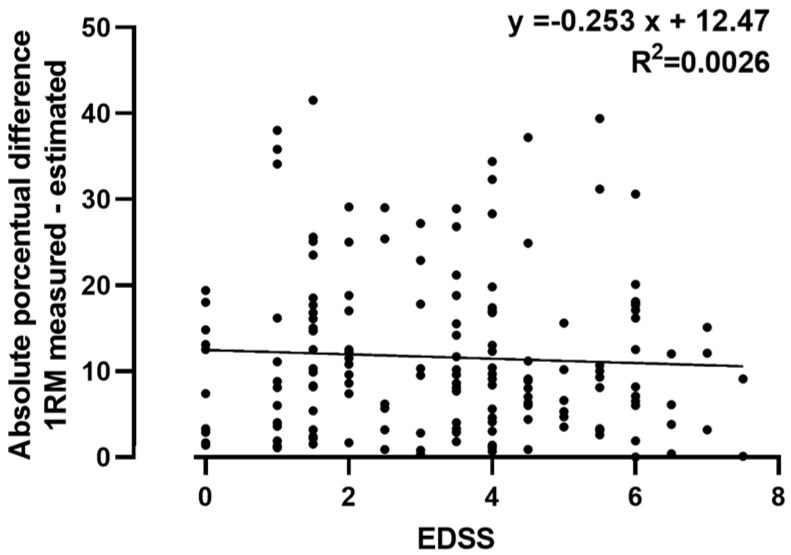
Correlation EDSS and absolute percentage difference in 1RM estimation.

**Table 1 medicina-58-00948-t001:** EDSS degrees and MS types of the participants.

		MS Type			
		RRMS (%)	PPMS (%)	SPMS (%)	∑ (%)
EDSS	Mild	74 (34)	7 (15)	3 (5)	84 (25)
	Moderate	107 (50)	26 (54)	24 (37)	157 (48)
	Severe	25 (11)	14 (29)	29 (45)	68 (21)
	Very Severe	10 (5)	1 (2)	8 (13)	19 (6)
	∑	216 (65.9)	48 (14.6)	64 (19.5)	328

MS—multiple sclerosis; RRMS—relapsing-remitting multiple sclerosis: PPMS—primary-progressive multiple sclerosis; SPMS—secondary-progressive multiple sclerosis; EDSS—Expanded Disability Status Scale.

**Table 2 medicina-58-00948-t002:** Variables of the sample grouped by sex.

	Male (123)					Female (205)						
	Mean		SD	Max	Min	Mean		SD	Max	Min	*p*	*d*
Age (years)	46.2	±	11.1	74.5	29	46.6	±	11.4	74	20	0.784	0.036
BMI (Kg/m^2^)	24.8	±	3.4	36.7	17.9	24.3	±	3.8	36.3	16.5	0.221	0.139
Evolution years	10.4	±	8.1	34	0	10.8	±	8.1	36	0	0.700	0.049
EDSS	3.7	±	2.1	8	0	3.4	±	1.9	8.5	0	0.245	0.150
MVIC (kg)	98.5	±	31.2	200	24.8	65.4	±	19.7	117	19.2	<0.001	1.301
1RM (kg)	87.2	±	25.4	150	16	60.3	±	21.5	120	12	<0.001	1.147

BMI—body mass index; EDSS—Expanded Disability Status Scale; MVIC—maximum voluntary isometric contraction; 1RM—one-repetition maximum; SD—standard deviation; Max, Min—highest and lowest values, respectively, of the subsamples; *p*—*p* value; d—Cohen’s d effect size.

**Table 3 medicina-58-00948-t003:** Reliability of isometric force.

	MVIC (kg)	Mean	SD	ICC	Confidence Interval 95%		CV %	SEM (kg)	SEM%	MDC (kg)	MDC %
All	Test 1	75.7	28.9	0.973	(0.967	0.978)	4.5	4.7	6.3	13.2	17.4
	Test 2	75.3	28.5								

MVIC—maximum voluntary isometric force; SD—standard deviation; ICC—intraclass correlation coefficient; CV%—coefficient of variation percentual; SEM—standard error of the measurement; MDC—minimum detectable change.

**Table 4 medicina-58-00948-t004:** Variables of the sample separated by training group and testing group.

	Training Group (164)					Testing Group (164)						
	Mean		SD	Max	Min	Mean		SD	Max	Min	*p*	*d*
Age (years)	47.4	±	11.1	74.5	24	45.5	±	11.4	73	20	0.180	0.169
BMI (kg/m^2^)	24.1	±	3.4	33.9	16.5	25.1	±	3.8	36.7	17.6	0.073	0.278
Evolution years	11	±	8	36	0	10	±	8	34	0	0.386	0.125
EDSS	3.6	±	2.1	8.5	0	3.5	±	2	8	0	0.731	0.049
MVIC (kg)	75.6	±	29.6	200	22	79.9	±	29	176.1	19.2	0.213	0.147
1RM (kg)	69	±	28.2	130	12	72	±	25	150	14	0.366	0.113

BMI—body mass index; EDSS—Expanded Disability Status Scale; MVIC—maximum voluntary isometric contraction; 1RM—one-repetition maximum; SD—standard deviation; Max, Min—highest and lowest values, respectively, of the subsamples; *p*—*p* value; d—Cohen’s d effect size.

**Table 5 medicina-58-00948-t005:** Equations of the linear regression of the total sample and by sex in training group.

	Slope	Intercept	R	R^2^	SEE (kg)
All samples	0.833	5.877	0.897	0.804	12.2
Male	0.718	17.64	0.854	0.729	14.8
Female	1.024	−6.67	0.895	0.801	9.4

R—correlation coefficient; R^2^—coefficient of determination; SEE—standard error of the estimate.

**Table 6 medicina-58-00948-t006:** MVIC and 1RM measured and estimated with two different regression functions in the testing group.

	MVIC (Kg)	1RM (Kg)	1RM_all (Kg)	1RM_sex (Kg)	*p*	d/η²	Δ1 (%)	Δ2 (%)	*p* (Δ1 vs. Δ2)	d
ALL	80.3 ± 28.3	72.6 ± 24.8	73.4 ± 25.3		0.463	0.028	12.7 ± 11.1			
Male	101.2 ± 28.7	87.8 ± 22.6	90.2 ± 25.2	91.4 ± 24.1	0.271	0.033	12.0 ± 9.4	12 ± 9.5	0.963	0
Female	67.5 ± 19.3	63.3 ± 21.3	61.9 ± 17.2	62.2 ± 19.8	0.731	0.001	10.8 ± 8.9	10.8 ± 8.7	0.589	0.022

MVIC—maximum voluntary isometric force; 1RM—one-repetition maximum; 1RM_all—one-repetition maximum estimated with the general equation; 1RM_sex—estimated with the equation by sexes; *p*—*p* value; d—Cohen’s d effect size; η²—partial eta-squared; Δ1—absolute percent error in the estimation with general equation; Δ2—absolute percent error in the estimation with equation by sex.

**Table 7 medicina-58-00948-t007:** Absolute percentage differences in the estimation between the 1RM value and 1RM_all by MS type.

MS Type								
RRMS		PPMS		SPMS				
Mean	S.D	Mean	S.D	Mean	S.D	*F*	*p*	η²
11.3 ± 9.5		13.9 ± 9.4		11.0 ± 9.8		0.791	0.456	0.011

MS—multiple sclerosis; RRMS—relapsing-remitting multiple sclerosis: PPMS—primary-progressive multiple sclerosis; SPMS—secondary-progressive multiple sclerosis; *p*—*p* value; η²—partial eta-squared; F—value of F.

## Data Availability

The datasets used and/or analyzed during the current study are available from the senior author on reasonable request, japazf@unileon.es.

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
