# Peer review of "Reliability of Isometric Muscle Strength Measurement and Its Accuracy Prediction of Maximal Dynamic Force in People with Multiple Sclerosis"

_medicina, 2022, doi:10.3390/medicina58070948_

Round 1

Reviewer 1 Report

Thanks for letting me read this paper. The main suggestions are as follows:

Introduction and method section.

1. The introduction and methods section does not make clear what contribution or significance several of the metrics measured in the study have to the subject of the study, for example, page 2, lines 94-97 suggest, "The purpose of this study was, in patients with MS, to analyze the absolute reliability of the assessment of the maximum isometric force of knee extension, and to analyze the correlation and the prognostic value that the MVIC has in the prediction of dynamic maximal strength (assessed with 1RM)." However, the text does not state what is the significance and value of analyzing the absolute reliability of knee extension maximum isometric strength assessment; similarly, the text does not state the mathematical or logical relationship between Isometric strength and 1RM, mentioned in 2.3 Measurement, and the purpose of the study, i.e., the absolute reliability of knee extension maximum isometric strength assessment.

2. The literature review section of the introduction is not well connected to the purpose of the study, that is, the contents of the literature review (page 2, lines 46-93) do not effectively support the purpose of the study (page 2, lines 94-97).

Discussion section

1. The authors should discuss the limitations of the study.

2. The authors should compare the study with other studies and should state what the strengths or what breakthroughs the study has compared to other studies and that the authors do not present the innovations of the study.

Conclusion section

1. In the conclusion section, the authors should deepen the importance of their results and future application scenarios, and any content of the study that requires future research to fill in the gaps, and the authors just repeated the results of the study.

Author Response

Author's Reply to the Review Report (Reviewer 1)

Before proceeding to indicate the changes made under your suggestion and to respond to your comments, we would like to thank you for your time and the comments you have made, which we believe improve our document.

Thank you very much.

REVIEWER'S REMARK: The introduction and methods section does not make clear what contribution or significance several of the metrics measured in the study have to the subject of the study, for example, page 2, lines 94-97 suggest, "The purpose of this study was, in patients with MS, to analyze the absolute reliability of the assessment of the maximum isometric force of knee extension, and to analyze the correlation and the prognostic value that the MVIC has in the prediction of dynamic maximal strength (assessed with 1RM)." However, the text does not state what is the significance and value of analyzing the absolute reliability of knee extension maximum isometric strength assessment; similarly, the text does not state the mathematical or logical relationship between Isometric strength and 1RM, mentioned in 2.3 Measurement”, and the purpose of the study, i.e., the absolute reliability of knee extension maximum isometric strength assessment.

Authors' response:

In the introduction we indicated that patients with MS often present muscle strength impairment, especially in the lower extremities (lin. 59-61), which can affect their level of dependence. We have also emphasized that especially in recent years, personalized strength training is recommended for these patients, programming loads based on the maximum dynamic strength (1RM) of each patient (lin. 62, 76-79). But still some professionals have some fear to evaluate the 1RM in these patients for fear of fatigue or injury (lin 79-82). Previously some studies have programmed workload based on maximal isometric strength (MVIC) or have used MVIC as an outcome variable of strength training (lin. 82-84). For this reason it is important to know the standard error of the measurement (SEM) and the minimum detectable change MCD, to better assess the effectiveness of the intervention through training.

To point out more emphatically the relevance of the study of the repeatability of the MVIC  assessment, clear as suggested by the reviewer, we have added in the introduction a text that in the original paper was in the discussion section:

“When a result variable is used (for example, the MVIC) to assess the effect of an intervention (for example, exercise), it is important to determine the degree of agreement between the values ​​obtained in a repeated evaluation in the same subject. This allows us to obtain the SEM and MDC of the method, so that we can make a more accurate interpretation of our results, because if the magnitude of the change experienced as a consequence of the intervention is greater than the MDC, we will have greater certainty that the change is not due to a lack of precision in the method used.

We have also mentioned the usefulness of knowing the correlation between 1RM and MVIC will allow us to make a certain functional comparison of published studies in which patients have had 1RM determined, with patients in published studies in which MVIC has been evaluated, (lin. 102-103).

Table 5 shows the parameters that define the linear regression function between MVIC (independent variable) and 1RM (dependent variable) obtained in the entire training group, 1RM= (5.877 x MVCI)+ 0.833), and also that obtained in the men of the training group (1RM= (17.64 x MVCI) + 0.718), and in women of the same group (1RM= (-6.67 x MVCI) + 1.024). (lin 263-272).

REVIEWER'S REMARK: The literature review section of the introduction is not well connected to the purpose of the study, that is, the contents of the literature review (page 2, lines 46-93) do not effectively support the purpose of the study (page 2, lines 94-97).

Authors' response:

We have pointed out, without emphasizing too much (so as not to be pedantic), that there are no studies that have analyzed the repeatability of MVCI assessment (lin 93-95), by writing the paragraph as follows:

“Isometric maximal strength has high test-retest reliability in different healthy populations of varied physical condition [24–27], but in pwMS this relationship has not yet been published in widely distributed scientific journals”

REVIEWER'S REMARK: The authors should discuss the limitations of the study.

Authors' response:

The strengths, practical applications and limitations of the study have been expanded and placed at the end of the discussion, by writing the paragraph as follows:

REVIEWER'S REMARK: The authors should compare the study with other studies and should state what the strengths or what breakthroughs the study has compared to other studies and that the authors do not present the innovations of the study.

Authors' response:

The fact that we have not compared our results with those of other studies is because, as we have indicated in a previous answer and in the document, there are no published studies on the repeatability of MVCI in patients with MS, nor correlation studies between MVCI and 1RM.

REVIEWER'S REMARK: In the conclusion section, the authors should deepen the importance of their results and future application scenarios, and any content of the study that requires future research to fill in the gaps, and the authors just repeated the results of the study.

Authors' response:

In the paragraphs preceding the conclusions, we have pointed out the main practical utilities of the study, as well as the main limitations and strengths. Writing the paragraph as follows:

However, there are some limitations to this study. First, is that the isometric force produced is related to the joint angle of the exercise [69,70]. Therefore, this extrapolation is valid only when the knee is flexed at around 95º and the hip at 110º. The results for weight machines with different angles between their levers can produce greater errors when applying the regression equation of this study.

Moreover, cross-sectional observational studies, such as the present study, do not report causal relationships or the effects of interventions. Therefore, we do not know whether the correlation found between MVIC and 1RM would be maintained or not after a period of strength training in these patients. Future studies will be needed to analyze the evolution of this correlation over a period of training. It will also be interesting to analyze and compare the strength values and correlations between these manifestations of muscle performance in patients with similar neurological damage determined by magnetic resonance imaging and functional magnetic resonance imaging.

We sincerely thank you for your contributions and hope that we have responded to your observations.

Thank you very much.

Reviewer 2 Report

1. Title.

1.1 The title is missing the article type and where the study was done. Multicentric?

2. Methods.

2.1 Who diagnosed MS cases? Were at least two board-certified neurologists?

2.2 What were the variables assessed in the logistic regression

2.3 How was calculated the power of the study?

3. Results.

3.1 Significant results should be marked with [*]

3.2 All "p" presented in this paper are statistical symbols and should be converted to italics

4. Discussion

4.1 The limitation of observational studies should be highlighted

The idea of the manuscript is not correlating with neurophysiology. First, the degree of neurologic damage is correlated with the severity of the cases. Second, to assess the correlation between muscle strength and maximal dynamic force in MS all the individuals should have the same area damaged calculated by fMRI. In the literature, we can find these calculators and some are provided for free. It is advised to the authors include this information in the study.

Could authors provide the spreadsheet database?

Author Response

Author's Reply to the Review Report (Reviewer 2)

We would like to express our deep appreciation to the reviewer for his time and for the corrections and observations he has made.

The observations that we have been able to address have been incorporated into the text. There are other observations that we cannot incorporate due to the methodology of our study or to methodologies that are not beyond our possibilities, which we also discussed with the reviewer.

Thank you very much.

REVIEWER'S REMARK: 1.1. The title is missing the article type and where the study was done. Multicentric?

Authors' response:

The study was not a multicenter study. The evaluations were all performed in the same laboratory and by the same investigators. The patients did live on a regular basis in different cities in our region. Specifically, they came from 9 associations: Leon, Zamora, Valladolid, Palencia, Palencia, Burgos, Miranda de Ebro, Ponferrada, Segovia and Soria. (We have added a joint text in response to this comment and comment 2.1).

REVIEWER'S REMARK: 2.1. Who diagnosed MS cases? Were at least two board-certified neurologists?

Authors' response:

All patients belonging to the associations of multiple sclerosis patients are diagnosed and treated by neurology specialists working in the Public Health Service of the Junta de Castilla y León, in the Neurology Services of the referral hospitals of each of the patient associations.

To make it clearer for readers, we have added the following text in response to the reviewer's two comments:

“This observational cross-sectional study was part of a strength training service for PwMS, which was performed by the University of León with the support of the Regional Ministry of Health of the of the Government of Castilla y León (Spain). All 328 participants (123 men and 205 women) were people affected by multiple sclerosis belonging to one of the nine MS patient associations of Castilla y León, and diagnosed by the neurology medical services of the Regional Health System of each of the cities where the associations are located. All evaluations were conducted by the same researchers from the Exercise Physiology Laboratory at the University of Leon, between March 2019 and March 2022.”

REVIEWER'S REMARK: 2.2 What were the variables assessed in the logistic regression.

Authors' response:

As indicated in the statistical methodology section, we did not perform logistic regression analysis, but simple linear regression to evaluate the prediction of 1RM from isometric strength, which was one of our objectives. In any case, neither age nor time of evolution was significantly involved in this relationship.

REVIEWER'S REMARK: 2.3 How was calculated the power of the study?

Authors' response:

We have added in the statistical procedure, how the minimum size needed for our study was calculated, and the post hoc power found with our study sample once we found the correlation between the predictor variable and the outcome variable. The text added is:

G*Power software (version 3.1.9.7) was used to calculate the minimum required sample size and the calculated power. To detect a minimum effect size of 0.2, with an alpha error of 0.05 and a power of 0.95 with two tails using a single predictor, a minimum sample size of 67 was calculated. After finding the correlation between the predictor variable (isometric strength) and the outcome variable (1RM) obtained in half of our sample, the post hoc power calculated with the G*Power software (version 3.1.9.7) was 1.0 (100%).

REVIEWER'S REMARK: 3.1 Significant results should be marked with [*]

Authors' response:

As indicated. In the articles of the journal Medicina, the statistical significance by "*" is reserved to be indicated in the graphs and figures. By noting in the statistical methodology that the significance level was set at p <0.05 and showing the p-value, we believe that the p-value is more informative than the categorization of significance and avoids repetition. However, if the journal policy is changed, we can indicate this by "*".

REVIEWER'S REMARK: 3.2 All "p" presented in this paper are statistical symbols and should be converted to italics

Authors' response:

The p-value abbreviations have been corrected and are now shown in italics.

REVIEWER'S REMARK: 4.1 The limitation of observational studies should be highlighted

 Authors' response:

We have expanded the wording of the study strengths and limitations, writing the paragraph as follows:

However, there are some limitations to this study. First, is that the isometric force produced is related to the joint angle of the exercise [69,70]. Therefore, this extrapolation is valid only when the knee is flexed at around 95º and the hip at 110º. The results for weight machines with different angles between their levers can produce greater errors when applying the regression equation of this study.

Moreover, cross-sectional observational studies, such as the present study, do not report causal relationships or the effects of interventions. Therefore, we do not know whether the correlation found between MVIC and 1RM would be maintained or not after a period of strength training in these patients. Future studies will be needed to analyze the evolution of this correlation over a period of training. It will also be interesting to analyze and compare the strength values and correlations between these manifestations of muscle performance in patients with similar neurological damage determined by magnetic resonance imaging and functional magnetic resonance imaging.

REVIEWER'S REMARK: The idea of the manuscript is not correlating with neurophysiology.  First, the degree of neurologic damage is correlated with the severity of the cases.

Authors' response:

We understand the reviewer's concern in the following comments.

But we would like to emphasize that our study did not look at the severity of the disease (aggressiveness, speed of progression, degree of response to treatment, importance of the neurological functions affected, importance of the affected functions on quality of life or functional independence...).

The EDSS is not synonymous with the severity of the disease, but only, to some extent, with the degree of neurological involvement. It is the scale most commonly used by neurologists who treat and follow these patients. This scale has criticisms, as for example the ability to ambulate has an important weight in the final score. It does not have the same weight in the final EDSS score, for example, the impairment of sphincter control as the ambulatory distance that the patient is able to perform.

We have studied isometric and dynamic strength (the patient does not have to move but to oppose or overcome a resistance). And there is not (in our experience, published experience) a relationship of a minimum acceptable degree, between the ambulatory capacity and the capacity to generate force. In ambulation, not only strength is involved, but also balance (not very dependent on muscle strength), coordination in the recruitment of agonist muscles and inhibition of antagonist muscles, the degree of spasticity, etc. ..... Thus, the degree of EDSS and strength present a poor correlation.

There is no severity scale that is generally used by all neurologists. There are several proposals for the analysis of severity (MS Severity Score: the decile of the EDSS within the range of patients who have had the disease for the same disease duration; Patient-Determined Disease Step (PDDS),...) but it has not been our purpose to relate or compare the strength or the repetitiveness of the same according to the degree of severity, we have only analyzed based on the degree of EDSS and what we found is what it usually is.

REVIEWER'S REMARK: Second, to assess the correlation between muscle strength and maximal dynamic force in MS all the individuals should have the same area damaged calculated by fMRI. In the literature, we can find these calculators and some are provided for free. It is advised to the authors include this information in the study.

Authors' response:

The reviewer also rightly points out that a study comparing damage observed by functional MRI (fMRI) scanning and strength would be very interesting. However, the availability of fMRI is not frequent, and the percentage of patients who have undergone this type of scan is minimal.

REVIEWER'S REMARK: Could authors provide the spreadsheet database?

Authors' response:

We are pleased to attach the database for the exclusive use of the reviewer. The policy of the Health Administration, which bears part of the costs of this study, does not allow us to provide public access to the data. In fact, we have to communicate individually to whom the data of the patients and of this study are provided.

Therefore, we would ask the reviewer, once the evaluation process of this article is finished (whether it is accepted or rejected), to kindly send an email to the senior investigator of this study, his or her data (name, affiliation, email and, if feasible, a telephone number), in order to meet the legal requirement that we have to register in the Regional Health Service of our Regional Government.

We understand that this type of study seeks a practical utility and does not have the interest of other studies with more resources. In fact, our study is intended to be published in a section of the journal of Sports Medicine, which is usually more useful for rehabilitation physicians, physiotherapists or physical educators.

We sincerely thank you for your contributions and hope that we have responded, to the best of our ability, to your observations.

Round 2

Reviewer 1 Report

The authors have done a good job of revising the paper.

Author Response

Many thanks for your time and orientations.

Reviewer 2 Report

1. The title is advised to provide the type of the study and the study location.

Author Response

We appreciate the time of the reviewer and have attended with interest to their observations. 

However, in relation to the second round:

1st We do not understand that it worsens your assessment of the English editing. In fact, we have not sent before the response to the second round, because we have transferred to MDPI's English pre-edit services, the reviewer's evaluation in relation to the English editing. Given that MDPI's translation service was the one who proofread the document, and we had also enclosed the certification of the translation service with the first submission.

2º Despite having taken into account all the observations of reviewer 2, incorporating them into the document, we do not understand that nothing has changed in his assessment (?), despite having provided him with the basis with the data that supports our work. He points out that everything should be improved and the only suggestion made to us is to add the type of study and the place where it was performed (?).

3º The generic observations made by reviewer 2 after the first round were answered with kindness and also with contrasted knowledge and proven experience in patients with Multiple Sclerosis, and in the object of study. We reasoned that these observations were decontextualized from the object of the study, from the field of rehabilitation and outside the possibility of common use by the vast majority of neurology services in the world (functional magnetic resonance imaging).

4º The only suggestion made is that we add the type of study and the place where the study was performed. Reviewing the titles of the publications in PubMed, we observe that we do not believe it is necessary, nor pertinent, to indicate in the title that it is a cross-sectional study. Nor is it necessary to indicate in the title the geographical location where the study was carried out, which is indicated in the chapter on methodology, which is where it is indicated in the vast majority of the articles published.

We can add little to the observations of reviewer 2 in the second round, except to respectfully state that we do not share his assessment.

However, we would like to thank you once again for your time, corrections and comments. 

Round 3

Reviewer 2 Report

The REVIEWER would discuss some points described by the authors.

1. According to the majority of guidelines regarding observational studies, the title should describe the type of study. The inclusion of the “localization where the study was done” is optional but some journals recommend it.

https://www.equator-network.org/

https://pubmed.ncbi.nlm.nih.gov/?term=multiple+sclerosis&sort=date&filter=pubt.clinicalstudy

Kearns PKA, Martin SJ, Chang J, Meijboom R, York EN, Chen Y, Weaver C, Stenson A, Hafezi K, Thomson S, Freyer E, Murphy L, Harroud A, Foley P, Hunt D, McLeod M, O'Riordan J, Carod-Artal FJ, MacDougall NJJ, Baranzini SE, Waldman AD, Connick P, Chandran S. FutureMS cohort profile: a Scottish multicentre inception cohort study of relapsing-remitting multiple sclerosis. BMJ Open. 2022 Jun 29;12(6):e058506. DOI: 10.1136/bmjopen-2021-058506. PMID: 35768080.

Almatrafi YM, Babakkor MA, Irfan M, Samkari ET, Alzahrani WM, Mohorjy DK, Zahoor S, Farooq MT, Sajid Jehangir HM. Efficacy and safety of rituximab in patients with multiple sclerosis: An observational study at a tertiary center in Makkah, Saudi Arabia. Neurosciences (Riyadh). 2022 Apr;27(2):65-70. DOI: 10.17712/nsj.2022.2.20210122. PMID: 35477910.

Bjornevik K, Cortese M, Healy BC, Kuhle J, Mina MJ, Leng Y, Elledge SJ, Niebuhr DW, Scher AI, Munger KL, Ascherio A. Longitudinal analysis reveals high prevalence of Epstein-Barr virus associated with multiple sclerosis. Science. 2022 Jan 21;375(6578):296-301. DOI: 10.1126/science.abj8222. Epub 2022 Jan 13. PMID: 35025605.

2. The reviewer does not believe that a quantification of the neuronal damage would not impact the quality of the study. Based on the pathophysiology of this disease, pwMS can have distinct areas and local damage. Therefore, the inclusion of specific area damage could standardize the data. Moreover, this turns the article reproducible. fMRI is only one of the possible ideas. There are studies with only the quantification of demyelination in MRI.  

Crinion J, Holland AL, Copland DA, Thompson CK, Hillis AE. Neuroimaging in aphasia treatment research: quantifying brain lesions after stroke. Neuroimage. 2013 Jun;73:208-14. DOI: 10.1016/j.neuroimage.2012.07.044. Epub 2012 Jul 27. PMID: 22846659; PMCID: PMC3534842.

Lipp I, Jones DK, Bells S, Sgarlata E, Foster C, Stickland R, Davidson AE, Tallantyre EC, Robertson NP, Wise RG, Tomassini V. Comparing MRI metrics to quantify white matter microstructural damage in multiple sclerosis. Hum Brain Mapp. 2019 Jul;40(10):2917-2932. DOI: 10.1002/hbm.24568. Epub 2019 Mar 19. PMID: 30891838; PMCID: PMC6563497.

https://www.ncbi.nlm.nih.gov/pmc/articles/PMC3534842/

3. Also, the REVIEWER believes that the questions were not decontextualized. The REVIEWER attempted to improve the manuscript provided by the authors. The questions were regarding title, methodology, statistics, limitations, data normalization and standardization, and spreadsheet data (for further evaluation). All of this information can be found in the STROBE checklist.

Cuschieri S. The STROBE guidelines. Saudi J Anaesth. 2019 Apr;13(Suppl 1):S31-S34. DOI: 10.4103/sja.SJA_543_18. PMID: 30930717; PMCID: PMC6398292.

Joanna Briggs Institute, Joanna Briggs Institute. Checklist for analytical cross-sectional studies. Adelaide: The Joanna Briggs Institute. 2017 Nov 6;7.

Barends E. A Reader’s Guide to Evidence-Based Management. Controlling & Management Review. 2016;1(60):36-40.

I have no further observations.